# SurroundDepth: Entangling Surrounding Views for Self-Supervised Multi-Camera Depth Estimation

**Yi Wei[1,2]\*, Linqing Zhao[3]\*, Wenzhao Zheng [1,2], Zheng Zhu [4], Yongming Rao[1,2],**

**Guan Huang [4], Jiwen Lu[1,2]†, Jie Zhou[1,2]**

[1]Beijing National Research Center for Information Science and Technology, China
[2]Department of Automation, Tsinghua University, China
[3]School of Electrical and Information Engineering, Tianjin University, China
[4]PhiGent Robotics

**Abstract:** Depth estimation from images serves as the fundamental step of 3D perception for autonomous driving and is an economical alternative to expensive depth sensors like LiDAR. The temporal photometric constraints enables self-supervised depth estimation without labels, further facilitating its application. However, most existing methods predict the depth solely based on each monocular image and ignore the correlations among multiple surrounding cameras, which are typically available for modern self-driving vehicles. In this paper, we propose a SurroundDepth method to incorporate the information from multiple surrounding views to predict depth maps across cameras. Specifically, we employ a joint network to process all the surrounding views and propose a cross-view transformer to effectively fuse the information from multiple views. We apply cross-view self-attention to efficiently enable the global interactions between multi-camera feature maps. Different from self-supervised monocular depth estimation, we are able to predict real-world scales given multi-camera extrinsic matrices. To achieve this goal, we adopt the two-frame structure-from-motion to extract scale-aware pseudo depths to pretrain the models. Further, instead of predicting the ego-motion of each individual camera, we estimate a universal ego-motion of the vehicle and transfer it to each view to achieve multi-view ego-motion consistency. In experiments, our method achieves the state-of-the-art performance on the challenging multi-camera depth estimation datasets DDAD and nuScenes. Code is available at https://github.com/weiyithu/SurroundDepth.

**Keywords:** Self-supervised depth estimation, Multi-camera perception, Structure-from-motion

## 1 Introduction

Recent years have witnessed the rapid development of autonomous driving. Instead of relying on expensive depth sensors like LiDAR to perform extract structural information, 3D perception from cameras has become a promising approach and potential alternative due to its semantic richness and economy. Acting as a bridge between the input 2D image and the real 3D environment, depth estimation has a crucial influence on the downstream 3D understanding and receives increasing attention [1, 2, 3, 4, 5, 6, 7, 8, 9, 10].

Due to the expensive cost of densely annotated depth maps, depth estimation is usually learned in a self-supervised manner. By simultaneously predicting the depth maps and the ego-motions of cameras, existing methods take advantage of the temporal photometric constraints between successive images as the supervision signal [11, 12, 13, 14, 15, 16]. Despite that modern self-driving cars

---

\*Equal contribution.
†Corresponding author.

6th Conference on Robot Learning (CoRL 2022), Auckland, New Zealand.

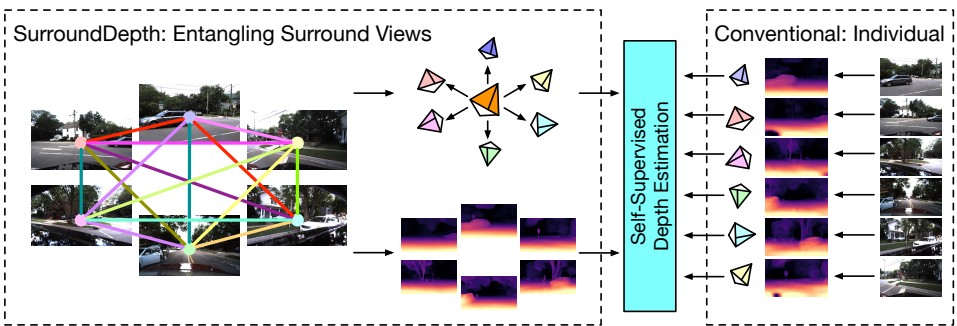

Figure 1: Comparison between SurroundDepth and self-supervised monocular depth estimation methods. Conventional methods [27, 28] predict depths and ego-motions of each view separately, which ignores the correlations between views. Our method incorporates the information across cameras and jointly process all surrounding views.

are usually equipped with multiple cameras to capture the full $360°$ view of the surrounding scene, most existing methods still focus on predicting depth maps from monocular images and ignore the correlations [17, 18, 19, 20, 21, 22, 23, 24, 25, 26] among the surrounding views [27, 28, 29, 30, 31].

In this paper, we propose SurroundDepth to process all the surrounding views jointly to produce high-quality depth maps across cameras. We first employ a shared encoder to extract high-level feature maps for each view and then propose a cross-view transformer to effectively fuse them. To efficiently allow multi-scale interactions between the features across cameras, we first reduce the resolution of feature maps with depthwise separable convolutions. Then we apply cross-view attention to integrate multi-camera features and use deconvolution to recover the original resolution. To alleviate the information loss induced by feature map downsampling, we add skip connections between individual and interacted feature maps. Finally, we use a shared decoder to attain the predicted depth maps. To recover the real-world scales, we propose Structure-from-Motion (SfM) pretraining and joint pose estimation to predict scale-aware results. In detail, we pretrain the depth estimation model with scale-aware pseudo depths derived from SfM [32]. Joint pose estimation is designed to achieve consistent pose predictions between cameras. By explicitly leveraging extrinsic matrices, we are able to calculate the individual ego-motion of each view from the predicted universal ego-motion.We conduct extensive experiments on the challenging multi-camera depth estimation datasets DDAD [27] and nuScenes [33]. Experimental results show that the cross-view feature interaction booost the performance and the proposed techniques can achieve real-scale depth estimation.

## 2   Related Work

**Self-supervised Monocular Depth Estimation:** Without available ground-truths, many approaches explore the routes of learning depths and motions simultaneously [11, 28, 12, 13, 14, 34, 15, 16, 35]. For monocular sequences, the geometric constraints are usually built on adjacent frames. Zhou *et al.* [11] built the problem as a task of view synthesis and trained two networks to separately predict poses and depths. Monodepth2 [28] further enhanced the quality of predictions by proposing the minimum re-projection loss, full-resolution multi-scale sampling, and auto-masking loss, which has been adopted in [36, 37]. Recently, FSM [29] extended self-supervised monocular depth estimation to full surrounding views by introducing both spatial and temporal contexts to enrich the supervision signals. Different from these monocular depth estimation methods, Our SurroundDepth captures cross-view interactions in surrounding views, which are important for fully understanding the environment with multiple cameras.

**Additional Supervision for Depth Estimation:** Recent approaches introduce additional supervision signals to strengthen the accuracy of depth estimation, such as optical flow [14, 38] and object motion [36, 16]. DispNet [39] was the first work to transfer information from synthetic stereo video datasets to the real-word depth estimation. Besides, Zheng *et al.*[40] proposed a two-module domain adaptive network with a generative adversarial loss to transfer knowledge from the synthetic domain. Some methods adopt auxiliary depth sensors to capture accurate depths, such as LiDAR, to assist depth estimation [41, 42]. To predict the depths with real-world scales, our method lever-

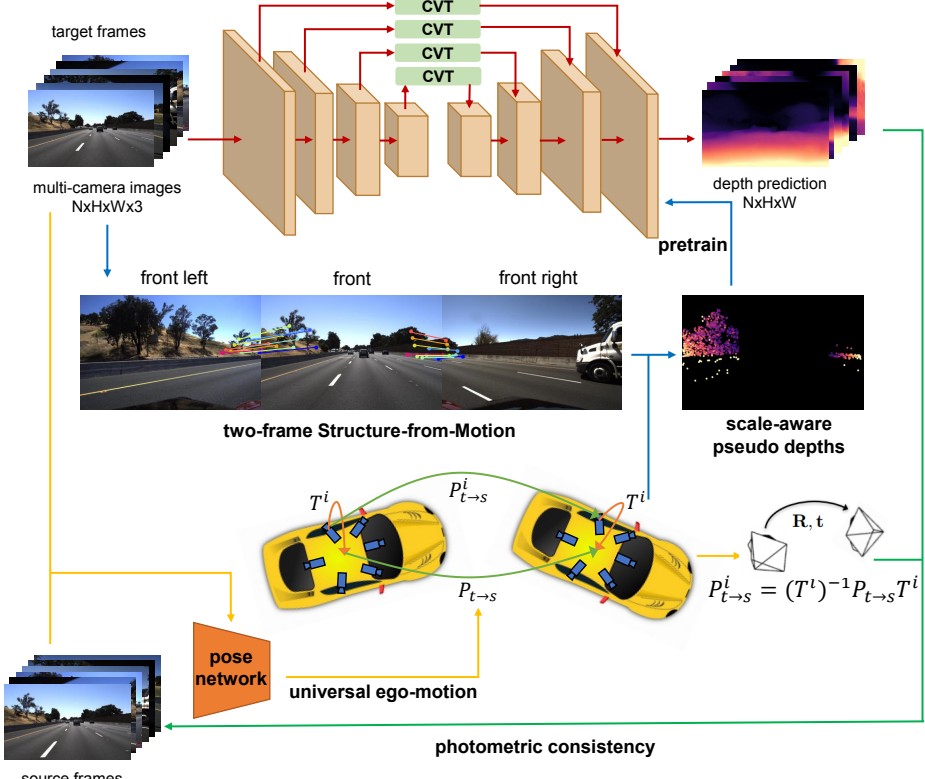

Figure 2: An overview of our SurroundDepth. We utilize encoder-decoder networks to predict depths. To entangle surrounding views, we propose the cross-view transformer (CVT) to fuse multi-camera features in a multi-scale fashion. Pretrained with the sparse pseudo depths generated by two-frame Structure-from-Motion, the depth model is able to learn the absolute scale of the real world. By explicitly introducing extrinsic matrices into pose estimation, we can predict multi-view consistent ego-motions and boost the performance of scale-aware depth estimation.

ages multi-camera extrinsic matrices and takes the sparse depths obtained from SfM to pretrain our network.

# 3 Approach

## 3.1 Problem Formulation

In the self-supervised depth and ego-motion settings, the depth network $F$ and pose network $G$ are optimized by minimizing a per-pixel photometric reprojection loss [43, 11]. This reconstruction-based self-supervision paradigm achieves great progress in monocular depth estimation methods [11, 12], and can be directly extended into multi-camera full surround depth estimation. The predicted depth maps and poses of a set of input surrounding samples $I = \{I^1, I^2, \cdots I^N\}$ can be written as:

$$D_t = \{D_t^{i}\}_{i=1}^{N} = \{F(I_t^{i})\}_{i=1}^{N}$$
$$P_{t \to s} = \{P_{t \to s}^i\}_{i=1}^{N} = \{G(I_t^i, I_s^i)\}_{i=1}^{N} \tag{1}$$

where subscript $s$ and $t$ mean the source and target frames. However, estimating full surrounding depths differs from monocular depth estimation in that there exists crucial correlations among the surrounding views, as shown in Figure 1. The overlaps between adjacent views connect all views into a full $360°$ view of the surroundings, which contains lots of beneficial knowledge and priors for

understanding the whole scene. Based on this fact, we build the joint models to predict depths and ego-motions instead of view-dependent estimations:

$$D_t = \{D_t{}^i\}_{i=1}^N = F(\{I_t{}^i\}_{i=1}^N)$$
$$P_{t \to s} = \{P_{t \to s}^i\}_{i=1}^N = G(\{(I_t^i, I_s^i)\}_{i=1}^N)$$

(2)

Profiting from the joint model, we can not only enable the cross-view information interactions to improve the performance of all views but also generate the universal ego-motion to produce scale-aware predictions with camera extrinsic matrices.

## 3.2 Overview

As illustrated in Figure 2, the network $F$ can be separated into three parts (*i.e.*, a shared encoder $E$, a shared decoder $D$, and several cross-view transformers (CVT)). Given a set of surrounding images, the encoder networks first extract their multi-scale representations in parallel. Different from existing methods that directly decode the learned features, we entangle the features from all views into an integrated feature at each scale, and further utilize multiple scale-specific CVT to perform cross-view self-attentions over all scales. By taking advantage of the powerful attention mechanism, CVT enables each element on the feature maps to propagate its information to other positions and absorb information from others at the same time. Finally, we separate the post-interactive features back to $N$ views and send them to the decoder $D$.

Unlike monocular depth estimation, we are able to recover the real-world scale from camera extrinsic matrices. A straightforward method to leverage these camera extrinsic matrices is to embed them into spatial photometric loss between two adjacent views. However, we find that the depth network fails to learn the scale directly supervised by spatial photometric loss. To tackle this issue, we propose scale-aware SfM pretraining and joint pose estimation. Specifically, we use two-frame SfM to generate pseudo depths to pretrain the models. The pretrained depth network is able to learn the real-world scale. Moreover, the temporal ego-motions of $N$ cameras have explicit geometric constraints. Instead of using a consistency loss, we estimate the universal pose of the vehicle and calculate the ego-motion of each view according to their extrinsic matrices.

## 3.3 Cross-View Transformer

With the multi-scale features extracted from all surrounding views, we replace the skip connections between the encoder and decoder with our proposed cross-view transformer (CVT), as shown in Figure 3. Let $X_k \in \mathbb{R}^{N \times H_k \times W_k \times d_k}$ be the feature maps obtained from the $k$-th scale, and we first supplement each feature with three unique learnable positional encodings ($PE_N, PE_H, PE_W$), which represent the view-wise, row-wise and column-wise index, respectively.

Then we build $Z$ cross-view self-attention layers to perform the cross-view information exchanging, whose computation cost is $\mathcal{O}(H_k^2 W_k^2 d_k)$ for each layer. However, the cost would be too enormous to accept when processing larger feature maps from the lower layers of the encoder. To avoid the huge cost caused by the quadratic term (*i.e.*, $H_k^2 W_k^2$), we place a depthwise separable convolution (DS-Conv) [44, 45] before attention layers to first summarize the large feature maps into lower-resolution ones with same channel numbers, *i.e.*, $X'_k \in \mathbb{R}^{N \times h_k \times w_k \times d_k}$. With much less computation than a standard convolution, the DS-Conv is widely adopted to balance the trade-off between efficiency and performance. To create the pathway across the features of surrounding views, we flatten the feature maps into an unified sequence, which includes the elements from all views, *i.e.*, $N \times h_k \times w_k$ elements in total. We develop three linear layers to obtain the query, key, and value vectors from $X'_k + PE_N + PE_H + PE_W$, and then we split the features into multiple groups along the channel dimension for multi-head self-attention to enrich the feature diversity. The output features can be represented as:

$$O_i^{out} = \text{Softmax}(K_i^T Q_i / \sqrt{d_k})V_i$$
$$X^{out} = \text{Concat}(O_1^{out}, \dots, O_M^{out})$$

(3)

where $M$ is the number of feature groups, and $K_i, Q_i, V_i$ denote the $i$-th feature group of the key, query, and value features, respectively. Benefiting from this attention unit, we ensure that every element on $X'_k$ not only interacts with intra-view features for better understanding of the current frame but also receives inter-view contexts for comprehensive cognition of the scene. To make

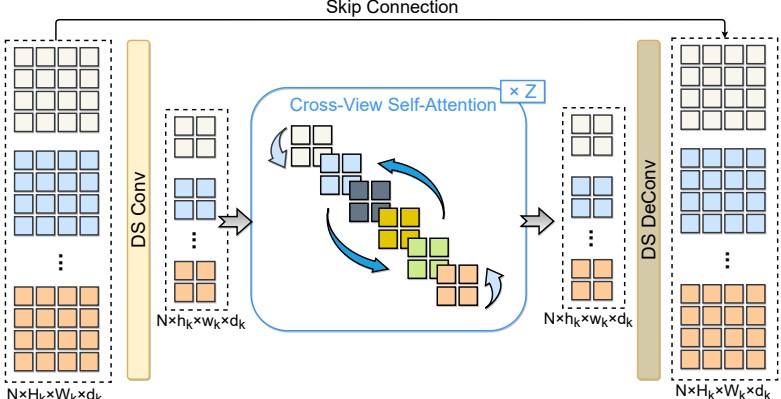

Figure 3: The proposed cross-view transformer. We first use a depthwise separable convolution (DS-Conv) layer to summarize the multi-view features into compact representations. Then we build $Z$ cross-view self-attention layers to fully exchange the flattened multi-view features. After the cross-view interactions, we use a DS-Deconv layer to restore the resolutions of multi-view features. At last, we construct a skip connection to combine the input and restored multi-view features.

cross-view information fully interactive, we stack such unit for $Z$ times in a cascade style. As an inverse process of flattening, we reshape the outputs of the $Z$-th attention layers back to the shape of $N \times h_k \times w_k \times d_k$, and further upsample the features to the original resolution of input features through a depthwise separable deconvolution. To alleviate the information loss caused by downsampling the feature maps and preserve the input details, we construct a skip connection to directly combine the input and output features:

$$D_k = X_k + X_k^{out} \tag{4}$$

By extending this strategy, features of all scales and all views can be fully propagated and updated from a global perspective.

### 3.4 Scale-aware Structure-from-Motion Pretraining

The aim of SfM pretraining is to explore the real-world scale from camera extrinsic matrices. A direct way to leverage extrinsic matrices is to use spatial photometric loss between two neighboring views, *i.e.*, warping $I_t^i$ to $I_t^j$:

$$p_t^{i \to j} = K^j (T^j)^{-1} T^i D_t^i (K^i)^{-1} p_t^i \tag{5}$$

where $K^i, T^i$ are the intrinsic and extrinsic matrices of $i$th camera. However, the overlap between $I^i$ and $I^j$ is relatively small, and the $p_t^{i \to j}$ is easy to be out of the image bounds if the scale of $D_t^i$ is far from the real-world scale. In this way, at the start of the training, the spatial photometric loss will be invalid and cannot supervise the depth network to learn the scale.

To tackle the problem, we first adopt SIFT [46] descriptors to extract correspondences. Then we compute scale-aware pseudo depths by triangulation with camera extrinsic matrices. Finally, we leverage these sparse pseudo depths along with the temporal photometric loss to pretrain the depth and pose networks. However, due to the small overlap and large view changes, the robustness and accuracy of descriptors decrease. To alleviate the issue, we only find corresponding points in a certain region instead of the whole image since we can roughly know the overlapping regions between two neighboring views. Specifically, in this work, we define left $\frac{1}{3}$ part of $I^i$ and right $\frac{1}{3}$ part of $I^{i+1}$ as the valid regions. Further, we leverage epipolar geometry to filter outliers. The epipolar line of point $q_m^i$ can be described as:

$$l_m = F_{i \to j} q_m^i$$
$$F_{i \to j} = (K^j)^{-T} [t]_\times R (K^i)^{-1} \tag{6}$$
$$[R|t] = (T^j)^{-1} T^i$$

where $F_{i \to j}$ is the fundamental matrix. If the distance between $q_m^j$ and epipolar line $l_m$ is larger than a threshold $\gamma$, we regard this correspondence pair as the noise. We provide an example in the supplementary materials.

| Method | Abs Rel | Sq Rel | RMSE | RMSE log | $\delta < 1.25$ | $\delta < 1.25^2$ | $\delta < 1.25^3$ |
|---|---|---|---|---|---|---|---|
| Monodepth2 [28] | 0.362 | 14.404 | 14.178 | 0.412 | 0.683 | 0.859 | 0.922 |
| PackNet-SfM [27] | 0.301 | 5.339 | 14.115 | 0.395 | 0.624 | 0.828 | 0.908 |
| Monodepth2 -M | 0.217 | 3.641 | 12.962 | 0.323 | 0.699 | 0.877 | 0.939 |
| PackNet-SfM -M | 0.234 | 3.802 | 13.253 | 0.331 | 0.672 | 0.860 | 0.931 |
| FSM [29] | 0.202 | - | - | - | - | - | - |
| FSM* [29] | 0.229 | 4.589 | 13.520 | 0.327 | 0.677 | 0.867 | 0.936 |
| SurroundDepth | **0.200** | **3.392** | **12.270** | **0.301** | **0.740** | **0.894** | **0.947** |

Table 1: Comparisons for self-supervised multi-camera depth estimation on DDAD dataset [27]. All methods are trained and tested with the same experimental settings. The results are averaged over all views with median-scaling at test time. *-M* indicates occlusion masking. * indicates our implementation.

| Method | Abs Rel | Sq Rel | RMSE | RMSE log | $\delta < 1.25$ | $\delta < 1.25^2$ | $\delta < 1.25^3$ |
|---|---|---|---|---|---|---|---|
| Monodepth2 [28] | 0.287 | 3.349 | 7.184 | 0.345 | 0.641 | 0.845 | 0.925 |
| PackNet-SfM [27] | 0.309 | 2.891 | 7.994 | 0.390 | 0.547 | 0.796 | 0.899 |
| FSM [29] | 0.299 | - | - | - | - | - | - |
| FSM* [29] | 0.334 | 2.845 | 7.786 | 0.406 | 0.508 | 0.761 | 0.894 |
| SurroundDepth | **0.245** | **3.067** | **6.835** | **0.321** | **0.719** | **0.878** | **0.935** |

Table 2: Comparisons for self-supervised multi-camera depth estimation on the nuScenes dataset [33]. All methods are trained and tested with the same experimental settings. The results are averaged over all views with median-scaling at test time. * indicates our implementation.

## 3.5 Joint Pose Estimation

An intuitive way of pose estimation is to predict the relative pose of each view separately, which can be represented by Equation 1. However, this strategy ignores the pose consistency between different views, which may lead to ineffective supervision signals. To maintain multi-view ego-motion consistency, we propose to decompose the camera pose estimation problem into two sub-problems: universal pose prediction and universal-to-local transformation. To obtain the universal pose $P$, we feed $N$ pairs of target and source images into the PoseNet $G$ at a time and take the average of extracted features before the decoder. The universal pose can be computed by:

$$\{h^i\}_{i=1}^N = G_E(\{(I_t^i, I_s^i)\}_{i=1}^N)$$
$$P_{t \to s} = G_D(\frac{1}{N} \sum_{i=1}^N h^{(i)})$$

(7)

where $G_E$ and $G_D$ denote the encoder and decoder of PoseNet, respectively. After obtaining the universal pose $P$, we can further transform it to each camera pose with known camera extrinsic matrices, which can be described as:

$$P_{t \to s}^i = (T^i)^{-1} P_{t \to s} T^i$$

(8)

where $T^i$ is the extrinsic matrix of $i$th camera and $P_{t \to s}^i$ is its ego-motion from target frame to source frame. By combining these two steps, we can obtain the theoretically consistent multi-camera poses, which will further improve the scale-aware depth estimation accuracy.

## 4 Experiments

### 4.1 Experimental Setup

We conduct experiments on both the Dense Depth for Automated Driving (DDAD) [27] and nuScenes datasets [33]. The basic backbones of depth and pose networks are the same as the Monodepth2 [28]. We employ ResNet34 with ImageNet [47] pretrained weight as the encoder for all experiments, including baseline methods. Further, surrounding cameras have different focal lengths and we refactor depth maps according to focal lengths following [48]. In each scale, we adopted $Z = 8$ transformer layers and all features were downsampled to 20×12 and 20×11 before cross-view attention on two datasets respectively. Since DDAD dataset has self-occlusion areas, following [29], we annotate occlusion masks manually and use them to reweight photometric loss. More implementation details and visualizations are in supplementary materials.

| spatial context | SfM pretrain | joint pose | Abs Rel | Sq Rel | RMSE | RMSE log | $\delta < 1.25$ | $\delta < 1.25^2$ | $\delta < 1.25^3$ |
|---|---|---|---|---|---|---|---|---|---|
| | | | 0.967 | 22.982 | 31.761 | 3.518 | 0.000 | 0.000 | 0.000 |
| ✓ | | | 0.978 | 24.062 | 31.980 | 3.749 | 0.000 | 0.000 | 0.000 |
| ✓ | ✓ | | 0.257 | 4.565 | 14.096 | 0.368 | 0.557 | 0.833 | 0.925 |
| ✓ | | ✓ | 0.881 | 19.499 | 29.552 | 2.224 | 0.000 | 0.001 | 0.002 |
| | ✓ | ✓ | 0.411 | 6.121 | 17.747 | 0.626 | 0.089 | 0.367 | 0.767 |
| ✓ | ✓ | ✓ | **0.208** | **3.371** | **12.977** | **0.330** | **0.693** | **0.871** | **0.934** |

Table 3: Quantitative results for scale-aware depth estimation on the DDAD dataset [27]. The scores are averaged over all views **without** median-scaling at test time. 'spatial context' denotes whether to use spatial context and multi-camera extrinsic matrices for training.

| spatial context | SfM pretrain | joint pose | Abs Rel | Sq Rel | RMSE | RMSE log | $\delta < 1.25$ | $\delta < 1.25^2$ | $\delta < 1.25^3$ |
|---|---|---|---|---|---|---|---|---|---|
| | | | 0.978 | 13.906 | 18.063 | 3.967 | 0.000 | 0.000 | 0.000 |
| ✓ | | | 0.970 | 13.702 | 17.931 | 3.654 | 0.000 | 0.000 | 0.001 |
| ✓ | ✓ | | 0.429 | 7.839 | 8.593 | 0.428 | 0.471 | 0.797 | 0.895 |
| ✓ | | ✓ | 0.969 | 13.661 | 17.896 | 3.620 | 0.000 | 0.000 | 0.001 |
| | ✓ | ✓ | 0.363 | **3.999** | 8.499 | 0.479 | 0.234 | 0.726 | 0.876 |
| ✓ | ✓ | ✓ | **0.280** | 4.401 | **7.467** | **0.364** | **0.661** | **0.844** | **0.917** |

Table 4: Quantitative results for scale-aware depth estimation on the nuScenes dataset [33]. The scores are averaged over all views **without** median-scaling at test time. 'spatial context' denotes whether to use spatial context and multi-camera extrinsic matrices for training.

## 4.2 The Benchmark for Multi-camera Depth Estimation

We compare our method with two existing state-of-the-art self-supervised monocular depth estimation methods (Monodepth2 [28] and PackNet-SfM [27]) and one multi-camera depth estimation method FSM [29]. Since FSM does not release code, its detailed experimental setting is unknown, such as evaluation, pretraining and hyperparameters. To fairly compare, we run all methods under the same training and evaluation settings. Specifically, the input images are downsampled to 640×384 and 640×352 resolutions. Depth ground truth is generated by projecting LiDAR point cloud to each view. Since Monodepth2 and PackNet-SfM are not designed for scale-aware depth estimation, we perform per-image median ground truth scaling [11] and post-processing [43] during evaluation. Moreover, SfM pretraining and joint pose estimation are designed for scale-aware depth estimation, we do not adopt them in this experiment. As shown in Tables 1 and 2, our method achieves state-of-the-art performance. Our method outperforms baseline method Monodepth2, which demonstrates that our proposed cross-view approach improves the results since we fully entangle surrounding views. We test inference time of one batch (6 images) on a single RTX 3090: Monodepth2 (0.028s), PackNet-SfM (0.471s), SurroundDepth (0.088s).

## 4.3 Scale-aware Self-supervised Depth Estimation

Different from monocular depth estimation, once we get the extrinsic matrices of each camera and use spatial context, the absolute scale of the world can be obtained. Tables 3 and 4 show the results without median-scaling at test time. We do not compare other methods since these methods cannot predict scale-aware depth maps. In contrast to the statement in [29], we find that the network directly utilizing spatial photometric loss cannot force the network to generate scale-aware depths. Without pretraining, the scale of initial depth is greatly different from the ground truth, which leads to invalid spatial photometric loss. With our proposed SfM pretraining, the depth network learns the absolute scale and spatial context becomes effective. The joint pose estimation further boosts the performance.

We further evaluate the depth consistency between surrounding views. Specifically, we project each view's depth map to its neighboring views with extrinsic matrices and calculate depth errors. Table 8 shows that the scale-ambiguous methods such as FSM and Monodepth2 cannot achieve multi-view consistency since the depth maps are scale-ambiguous but extrinsic matrices are scale-aware. With the cross-view interaction, we can further boost the consistency.

| skip | multi-scale | Abs Rel | Sq Rel | $\delta < 1.25$ |
|---|---|---|---|---|
| | | 0.217 | 3.641 | 0.699 |
| ✓ | | 0.209 | 3.385 | 0.704 |
| | ✓ | 0.222 | 3.763 | 0.695 |
| ✓ | ✓ | **0.200** | **3.392** | **0.740** |

Table 5: The ablation study for cross-view transformers. 'skip' and 'multi-scale' denote skip connection and multi-scale cross-view transformer.

| Pose type | Abs Rel | Sq Rel | $\delta < 1.25$ |
|---|---|---|---|
| separate | 0.257 | 4.565 | 0.557 |
| constraint | 0.254 | 4.390 | 0.576 |
| joint | **0.208** | **3.371** | **0.693** |
| ground truth | 0.210 | 4.164 | 0.729 |

Table 6: The ablation study for joint pose estimation. 'separate' indicates that we predict the pose of each view separately. 'constraint' further adds multi-camera consistency constraints.

| region mask | epipolar filter | Abs Rel | Sq Rel | $\delta < 1.25$ |
|---|---|---|---|---|
| | | 0.241 | 3.970 | 0.629 |
| ✓ | | 0.236 | 3.684 | 0.649 |
| ✓ | ✓ | **0.208** | **3.371** | **0.693** |

Table 7: The ablation study for SfM pretraining. 'region mask' indicates the region constraint for SIFT matching. 'epipolar filter' indicates the epipolar constraints to filter out noisy labels.

| scale-aware | CVT | Abs Rel | Sq Rel | $\delta < 1.25$ |
|---|---|---|---|---|
| | | 0.989 | 28.872 | 0.000 |
| ✓ | | 0.319 | 18.217 | 0.732 |
| ✓ | ✓ | **0.257** | **8.027** | **0.735** |

Table 8: Depth consistency evaluation. 'scale-aware' indicates the scale-aware model. 'CVT' means cross-view transformers.

## 4.4 Ablation Study

In this subsection, we conduct ablation studies to verify the effectiveness of each module in our framework. All experiments are conducted on DDAD dataset.

**Cross-view Transformer:** Table 5 shows that both skip connection and multi-scale formulation contribute to the final results. Skip connections are able to alleviate the information loss caused by the downsampling of feature maps. The multi-scale cross-view transformer helps the network to better learn both fine-grained and global features.

**Joint Pose Estimation:** Table 6 shows separate pose estimation cannot guarantee the multi-camera pose consistency. To provide a straightforward baseline (second line), we project the ego-motion of all views to the front view and add a loss between the projected and predicted ego-motion of the front view. However, this method gets marginal improvements. We note that our joint pose estimation method gets comparable performances with ground-truth pose experiment. This result indicates that ground truth ego-motion may have noise, even for carefully-calibrated public datasets. Thus, it is necessary to predict accurate scale-aware poses.

**Scale-aware SfM Pretraining:** Table 7 shows that region mask is able to help our model to learn real-world scale. If we do not set region mask for SIFT matching, we will get wrong correspondences and the depth network is not able to learn real-world scale. By filtering out the correspondences far from the epipolar lines, we will further boost the performance of the depth network.

## 5 Limitations

Although the proposed SurroundDepth can boost the multi-view depth consistency with scale-aware training strategy and cross-view transformers (as shown in Table 8), our method cannot theoretically guarantee this consistency. As future work, we will directly predict voxel occupancy of the 3D space instead of each view's depth maps.

## 6 Conclusion

In this paper, we propose SurroundDepth for self-supervised multi-camera depth estimation. The core insight of our method is to entangle multi-camera information and jointly process all surrounding views. The cross-view transformer is performed at multiple scales to incorporate multi-view features. To obtain scale-aware depth predictions, we present Structure-from-Motion pretraining and joint pose estimation to fully leverage multi-camera extrinsic matrices. Our method achieves state-of-the-art performance on the multi-camera depth estimation datasets.

**Acknowledgments**

This work was supported in part by the National Key Research and Development Program of China under Grant 2017YFA0700802, in part by the National Natural Science Foundation of China under Grant 62125603 and Grant U1813218, in part by a grant from the Beijing Academy of Artificial Intelligence (BAAI).

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
