# OpenReview forum: "SurroundDepth: Entangling Surrounding Views for Self-Supervised Multi-Camera Depth Estimation"
_robot-learning.org/CoRL/2022/Conference — CoRL 2022 Poster_

### Official Review · Reviewer_nGH3 · 2022-07-07

**Originality:** Good
**Technical Quality:** Good
**Clarity Of Presentation:** Good
**Impact:** 3

**Recommendation:**

Weak Accept: I recommend accepting the paper, but will not argue for my recommendation if the majority of other reviewers have a different opinion.

**Summary:**

This paper proposed a SurroundDepth module to predict the full surround depth maps from multiple cameras.
The major contribution includes:
* A novel cross-view transformer with cross-view self-attention to fuse the multi-view information and enable interactions.
* An SfM(structure-from-motion) pretraining mechanism is proposed to predict the metric scales of the depth.

**Issues:**

**Major Issues:**
* Since the FSM[29] is the only fair comparison baseline, could you use the same evaluation settings or at least illustrate the detailed difference in evaluation?
* The Transformers are generally computational heavy, could you further compare the computational cost such as the number of parameters, consumption of GPU-RAM, and training time.
**Minor Issues:**
* The KITTI Monocular/Stereo performance results are appreciated if possible.
* The limitation analysis could be more comprehensive.

**Quality Of The Limitations Section:**

Additional details required

**Reviewer Expertise:**

5: The reviewer is absolutely certain that the evaluation is correct and very familiar with the relevant literature

**Robotics Focus:**

Highly relevant to robotics but no hardware experiments

**Strengths And Weaknesses:**

**Strengths:**
* Although the full-surround depth prediction is not a new concept(e.g. FSM[29]), the author proposed a novel cross-view transformer and self-attention mechanism to better fuse the feature and enable interactions to improve the accuracy.
* The proposed Scale-aware SfM pretraining is effective to retrieve the metric scale for depth prediction.
**Weaknesses:**
* The experiment results in Table3 and 4 are inconsistent with paper FSM[29], it is better to use the same evaluation settings or at least illustrate the detailed difference in evaluation.
* It's better to also test on the KITTI dataset with the Stereo setting to better compare with the majority of the depth prediction methods.
* The comparison of test inference time with FSM[29] is missing, what is its inference time on the same hardware setting?
* It's better to also compare the number of parameters, consumption of GPU-RAM, and training time with the baseline methods.

**Summary Of Recommendation:**

The author introduced the Transformer into full-surround depth prediction task and achieved significant performance improvement according to the experiment section. However, the evaluation settings need to be further illustrated and the computational cost should be further analyzed.

Post-rebuttal:
(1)The evaluation comparison with FSM[29] should be updated.
(2)The proposed method consumes significantly more computation and memory while the performance improvement looks marginal.
(3)The introduction of Transformers to this task and the open-source code do contribute to the community, which is appreciated.
Based on the above considerations, my final rating is "weakly accept"

---

> ### Author Response · Authors · 2022-08-17
> **Comparison with FSM**
>
> Dear reviewer,
>
> Thanks for your valuable comments and appreciation of our method. Now I would like to talk about the comparison with FSM and this is a long story.
>
> In the last year, many companys ask us if we can provide a multi-camera depth estimation method since the only relevant method FSM does not release code. To reproduce FSM, we spent almost all of the winter vocation but we still could not reproduce the reported results. Besides us, many other researchers also cannot reproduce results (https://github.com/TRI-ML/packnet-sfm/issues).  As you know, if the first work in a new area does not provide code, it is difficult to follow and compare with this work, which is not a good thing for the community.
>
> Recently, the authors of FSM provide the inference model on DDAD dataset and the evaluation code is same with ours. However, the training code is still not available and we still cannot reproduce their method and compare training setting. To benefit the community, we have released our code and we hope our codebase can help other researchers.
>
> For a more fair comparison, we add the reported results in FSM paper. Since they only give Abs Rel result, we only compare on this metric:
>
> |method | type | dataset | Abs Rel |
> |:-------:|:-------:|:-------:|:-------:|
> |Ours| scale-ambiguous | DDAD | 0.200|
> |FSM| scale-ambiguous | DDAD | 0.202|
> |Ours| scale-aware | DDAD | 0.208|
> |FSM| scale-aware | DDAD | 0.201|
> |Ours| scale-ambiguous | nuScenes | 0.245|
> |FSM| scale-ambiguous | nuScenes | 0.299|
> |Ours| scale-aware | nuScenes | 0.280|
> |FSM| scale-aware | nuScenes | 0.297|

---

> > ### Comment · Reviewer_nGH3 · 2022-08-18
> > **Concerns on the significance**
> >
> > Dear Author:
> >
> > From the results provided, the proposed method requires much more computation time and GPU memory but the accuracy improvement is not significant in most cases. I doubt if it is necessary to introduce the complex model to this task. However, I still appreciate the contribution to the community by open-sourcing the first multi-camera depth solution.

---

> > > ### Author Response · Authors · 2022-08-19
> > > **About the significance**
> > >
> > > Dear reviewer:
> > >     Thanks for your appreciation ! We admit that our work may not be the most effective way for multi-camera depth estimation since transformers are computational heavy. However, we want to tell other researchers that using transformers can boost the performance and it is a right direction to integrate multi-view features . BEVFormer (ECCV 22') also employs transformer in their backbone. Their base method even cannot run on RTX 3090 and needs V100. We should notice that this is a new task and there is a long way to go. We hope our method can give inspiration to other researchers and our codebase can help this new area.

---

> > ### Comment · Reviewer_nGH3 · 2022-08-18
> > **Can you add these results to the main paper?**
> >
> > Dear Author:
> >
> > I hope these comparisons could be updated to the main paper.

---

> > > ### Author Response · Authors · 2022-08-19
> > > **Revision of the paper**
> > >
> > > Dear reviewer:
> > >
> > > Sure, we will add this table in the revised version. However, there is no space for a table and could we add it in the supplementary material ? We will also add one sentence in the main paper to tell readers supp has this comparison.

---

> > > > ### Comment · Reviewer_nGH3 · 2022-08-20
> > > > **Revision**
> > > >
> > > > Dear Author:
> > > >
> > > > For a fair comparison, I think you should replace the FSM results in table 1&2 with the ones from their original paper. The current result could be another row and marked as "our implementation".

---

> > > > > ### Author Response · Authors · 2022-08-21
> > > > > **Thanks**
> > > > >
> > > > > Dear reviewer:
> > > > >
> > > > > Thanks for your suggestion ! We will update these tables in our revised version.

---

> ### Author Response · Authors · 2022-08-18
> **Computational cost analysis**
>
> We analyse the computatioal cost of our Surrounddepth and baseline methods.
>
> training:
> | Method     | time | GPU-RAM |
> |:-------:|:-------:|:-------:|
> |Monodepth2| 17.1h | 10.5G |
> |FSM| 18.5h | 11.3G |
> |PackNet-SfM| 31.2h | 27.2G|
> |SurroundDepth| 23.2h | 18.9G |
>
> inference:
> | Method     | time | GPU-RAM |
> |:-------:|:-------:|:-------:|
> |Monodepth2| 0.028s| 2.9G |
> |FSM| 0.028s | 2.9G |
> |PackNet-SfM| 0.471s | 12.2G|
> |SurroundDepth| 0.088s | 3.2G |
>
> For training we use 8 RTX 3090 and for inference we use one RTX 3090 with one batch (6 images). We should notice that monodepth2 and FSM have the same inference results since their depth networks are same. We admit that the transformer will increase computation cost but we think that the increase is acceptable.

---

> > ### Comment · Reviewer_nGH3 · 2022-08-18
> > **Does Monodepth2 and PackNet-SfM only using single GPU?**
> >
> > Dear Author:
> >
> > Monodepth2 and PackNet-SfM only need a single GPU with 10-20h training time on the KITTI/Cityscapes dataset. So are they also using a single GPU and your model uses 8 GPU for training?

---

> > > ### Author Response · Authors · 2022-08-19
> > > **Training setting**
> > >
> > > Dear reviewer:
> > >
> > > For fair comparison, all methods are trained with 8 GPUs. Different with KITTI, our task has more training samples since the inputs have 6 views.

---

### Official Review · Reviewer_maRz · 2022-07-28

**Originality:** Very Good
**Technical Quality:** Very Good
**Clarity Of Presentation:** Excellent
**Impact:** 3

**Recommendation:**

Strong Accept: I recommend accepting the paper and will argue for my recommendation even if other reviewers hold a different opinion.

**Summary:**

This paper presents a novel method for multi-view depth prediction based on multiple color images without the usage of calibrated depth sensor (LIDAR). They propose extracting features with a simple backbone (ResNet34), which are combined with a Cross-View-Self-Attention method, where the final results are deconvoluted and combined with skip connections from the encoder to keep detail.
They show how to predict a scale for the depth estimations based on a nice pretraining on structure-from-motion data.
Furthermore, they predict the joint pose for all cameras, showing that they even can outmatch the error in the ground truth data.
They also show superior results in comparison to FSM, PackNet-SfM and Monodepth2.


**Issues:**


Not relevant anymore:
* more clearly show that your multi-view approach improves over using a non-multiview approach
* add a point cloud visualization to the supplementary

**Quality Of The Limitations Section:**

Limitations are addressed clearly

**Reviewer Expertise:**

3: The reviewer is fairly confident that the evaluation is correct

**Robotics Focus:**

Relevant but unlikely to deploy to hardware in near future

**Strengths And Weaknesses:**

Strengths:
* The paper is well written and easy to follow, something I do not say lightly
* I especially like the idea of using an attention model for incorporating the information from the different views
* great ablation, most questions I had were evaluated and proven to be improved

Weaknesses:
* the only thing I do not fully see proven, but it might be that I need a minor clarification, is that the method works better when it uses all images simultaneously. I feel these results are covered in table 8., but they are not fully discussed in 4.3. Can I get a clarification on this.
* one thing I particularly do not like is when depth images are solely represented as images and not as point clouds. It is nearly impossible to judge a depth image in 2D, anyone who claims they can judge that is lying. I would have love an image or even better, a video of a point cloud, with the different metrics turned on and off to immediately see the difference


**Summary Of Recommendation:**

I strongly recommend this paper for acceptance to CORL. They have proven that this paper deserves to be published in CORL.



Old Review before Discussion: I will be absolutely clear in my review. I think this paper deserves a strong accept. The idea is good the evaluation is great. I just have one question, can you prove that using multiple images with the cross-view-self-attention method improves the results? If that is already hidden in table 8. please tell me, and I will gladly change my review to strong accept. If not, please add these results, and I will also change my review.

---

> ### Author Response · Authors · 2022-08-16
> **About pre-rebuttal rating score "Strong Reject"**
>
> Dear reviewer,
>      Thanks for your valuable comments! We wonder if you make mistake about the pre-rebuttal rating score. We find that most of your comments are positive and you also say that "I think this paper deserves a strong accept", however, the rating score is "Strong Reject". We will respond  to your concerns in a few days.

---

> > ### Comment · Reviewer_maRz · 2022-08-16
> > **No mistakes made**
> >
> > Dear authors,
> >
> > there was no mistake made. As I add in my summary, I want to see that using multiple images actually improves the results, else, the central claim of your paper is not there, and without it, I can not accept this paper. I kind of suspect table 8 to contain this, but I would like to have confirmation and would like to stress that this point should be made more evident in the paper.
> >
> > I know this seems harsh, but you have to prove your main contribution.

---

> > > ### Author Response · Authors · 2022-08-17
> > > **The effectiveness of our cross-view approach**
> > >
> > > Dear reviewer,
> > >
> > > Thanks for your responce. Actually, Monodepth2 is the baseline of our method, which predicts depth images of each view independently and does not apply interaction between multi-camera features. Our method has the same backbone architecture with Monodepth2 and the difference is that Monodepth2 does not employ our cross-view techniques. From Table 1 and Table 2, we can see that our cross-view approach improves the results since we fully entangle surrounding views. We will emphasize this fact in our revised version. Besides the performance, Table 8 demonstrates that our approach can also improve the multi-view depth consistency of 6 cameras.

---

> > > > ### Comment · Reviewer_nGH3 · 2022-08-17
> > > > **Concerns on the baseline**
> > > >
> > > > Dear Author:
> > > >
> > > > I am concerned that Monodepth2 and PackNet-SfM are not a good baseline for this comparison. First, they are initially designed for monocular depth prediction, but I think at least stereo-based methods would be more appropriate in this scenario. Second, Monodepth2 and PackNet are currently not very strong baselines even in monocular settings. Could you add stronger baselines to make the comparison more convincing?

---

> > > > > ### Author Response · Authors · 2022-08-17
> > > > > **Discussion about baseline methods**
> > > > >
> > > > > Dear reviewer,
> > > > >
> > > > > Thanks for your comments. We chose monodepth2 and packnet-sfm as our baselines for the following reasons.
> > > > >
> > > > > 1. Monodepth2 (ICCV 19') is one of the most well-known and widely-used baselines in the depth estimation field, because its code is very clean, contains few tricks, and the performance can be reproduced. Specifically, Monodepth2 employs a pure self-supervised monocular depth estimation paradigm, which has been regarded as a classic framework. Due to these reasons, our method is built upon monodepth2 codebase. The basic backbone architecture of our method is same to Monodepth2 and the comparison with it can verify the effectiveness of the proposed cross-view techniques.
> > > > >
> > > > > 2.  Recent SOTA monocular depth estimation methods adopt Monodepth2's framework and add their new ideas (e.g., Discrete wavelet transform in WaveletMonodepth (CVPR 21'),  structure distillation in DistDepth (CVPR 22')) to this basic framework. Since our SurroundDepth focuses on integrating multi-camera information and is complementary to other monocular methods, these ideas can also be applied in our framework.  As one of SOTA method, PackNet-sfm (CVPR 20') extends Monodepth2 with a stronger backbone. It achieves top performance on DDAD (front view), so that we choose PackNet-sfm as a stronger baseline to demonstrate the effectiveness of our method. We believe that FSM only chooses Monodepth2 and PackNet-sfm as their baselines for the same reason.
> > > > >
> > > > > 3. We should notice that multi-camera depth estimation is a new but important setting for autonomous driving applications and there only exists one relevant method FSM.  It is also not fair to compare with deepMVS/stereo-based methods since these methods assume that multi-view images should have large overlaps but overlaps between surrounding views are very small (Figure 3 in supp). FSM also presents the same idea, and here we quote the statements in their paper
> > > > > "MVS-based methods proposed setting assumes a large collection of images surrounding a single object with known relative pose and large overlap for cost volume computation, and is thus very different from our setting – our architecture is designed to work with image sequences from any location and with arbitrarily small overlapping between cameras."
> > > > >
> > > > > Although we think we do not need to compare with other baselines, if the reviewer want us to compare with certain methods, please tell us these methods and we are willing to add the comparison with them.

---

> > > > ### Comment · Reviewer_maRz · 2022-08-18
> > > > **That clears things up**
> > > >
> > > > Thanks for this confirmation. This information should be stressed more in the paper. It maybe should be even already mentioned in the introduction.
> > > >
> > > > I will change my review to strong accept. Great work.

---

> > > > > ### Author Response · Authors · 2022-08-18
> > > > > **Thanks**
> > > > >
> > > > > Thanks for your appreciation of our work! We will update it in the final version.

---

> > > > > ### Author Response · Authors · 2022-08-19
> > > > > **About changing rating score**
> > > > >
> > > > > Dear reviewer:
> > > > >
> > > > > In instructions for reviewers (https://corl2022.org/instructions-for-reviewers/), we see that reviewers can update the score during the rebuttal period. So we wonder if you could change the rating score now. We know that it may not be polite to urge reviewers to change the score, but a “strong reject” gives us too much pressure and we hope the reviewer can comprehend us. We are looking forward to your feedback and will be happy to answer any further questions you may have.

---

> > > > > > ### Comment · Reviewer_maRz · 2022-08-19
> > > > > > **Changing rating score**
> > > > > >
> > > > > > I fully agree with that. I would love to change the rating score, but I can't find out how. The edit button doesn't allow me to edit the review.
> > > > > >
> > > > > > I will try to contact the AC, and maybe they have to allow us the edit the reviews.

---

> > > > > > > ### Author Response · Authors · 2022-08-19
> > > > > > > **Thanks**
> > > > > > >
> > > > > > > Dear reviewer:
> > > > > > >
> > > > > > > Thanks for your understanding ! Thank you very much !

---

### Official Review · Reviewer_RZGD · 2022-07-31

**Originality:** Very Good
**Technical Quality:** Very Good
**Clarity Of Presentation:** Very Good
**Impact:** 3

**Recommendation:**

Weak Accept: I recommend accepting the paper, but will not argue for my recommendation if the majority of other reviewers have a different opinion.

**Summary:**

This paper proposed a method that predicts high-quality depth maps across cameras by exploring correlations among surrounding views. They apply the cross-view attention to fuse the downsampled feature maps. They use structure-from-motion on adjacent frames to obtain the scale-aware pseudo depths. To exploit the pose consistency between views, they propose to predict the universal pose and then transform it to each local camera pose instead of predicting the pose of each view separately.


**Issues:**

Please address my concerns in the weaknesses.

**Quality Of The Limitations Section:**

Additional details required

**Reviewer Expertise:**

4: The reviewer is confident but not absolutely certain that the evaluation is correct

**Robotics Focus:**

Highly relevant to robotics but no hardware experiments

**Strengths And Weaknesses:**

Strength
This paper is well written and easy to follow. Their method shows good results on two datasets compared to the existing works. The experimental results show that they could effectively incorporate the information from multiple views. They provide insights to supervise the depth network to learn the real-world scale. The joint pose estimation proved to be having a positive effect on the performance. Also, the reported model’s inference time is fast.

Weakness
The reported scores of tables 1-4 are averaged over all views. Are the performance gains consistent on each separate view?
As shown in table 4, the setting of w/o spatial context might lead to worse results on the metrics of Sq Rel. Also, the performance gap between w/o spatial context and the whole setting is smaller than that on the DDAD dataset. The area of overlaps between adjacent views might be different in the different datasets. Does it bring some restrictions to this method?
FSM uses a more light-weight encoder(Resnet-18) compared to the ResNet-34 using here. It is good to see the information on the FSM’s inference time.
This work proposes to exploit information from multiple views to make predictions. In real applications, it is common to have extrinsic noises, one camera occluded, or camera time delay. It is good to see the robust analysis.


**Summary Of Recommendation:**

I think results and the techinical solution are convincing and interesting. I would recommend weak accept.

---

> ### Author Response · Authors · 2022-08-20
> **Restriction of the method**
>
> Dear reviewer:
>
> Thanks for your valuable comments !  Since we leverage overlapped areas to intergrate multi-view features in cross-view transformer and extract pseudo depths in SfM pretraining,  our method need multi-camera overlaps. As you can see, the performance gap between w/o spatial context on nuScenes dataset is smaller than that on the DDAD dataset. This experimental result shows that larger overlap is beneficial for our method.

---

> ### Author Response · Authors · 2022-08-20
> **FSM’s inference time**
>
> Dear reviewer:
>
> To fairly compare, the FSM experiments in Table 1-2 are conducted with backbone ResNet34, which is claimed in line 177. Since FSM has the same depth network with Monodepth2, these two methods have same inference time. We test inference time of one batch (6 images) on a single RTX 3090: Monodepth2, FSM (0.028s), PackNet-SfM (0.471s), SurroundDepth (0.088s). We admit that transformer will increase some inference time but we think the increment is acceptable.

---

> ### Author Response · Authors · 2022-08-22
> **Performance gains on each view**
>
> Dear reviewer:
>
> We find that our SurroundDepth surpass baseline methods on each view. We take monodepth2 as an example.
>
> Ours:
> | View| Abs Rel | Sq Rel | RMSE | RMSE log | delta < 1.25 | delta < 1.25^2 | delta < 1.25^3 |
> |:-------:|:-------:|:-------:|:-------:|:-------:|:-------:|:-------:|:-------:|
> | front |0.141  | 3.019  |13.472  |  0.228  |  0.832  |  0.941  |  0.974|
> |front left | 0.196  | 3.235  | 11.957  | 0.303  | 0.747  | 0.895  | 0.945|
> |back left| 0.212 | 3.560 | 11.709  | 0.312  | 0.727  | 0.885  |  0.940 |
> |front right| 0.222  | 3.561  |11.350  |  0.325  |  0.709  |  0.879 |   0.939|
> |back_right| 0.231  |  3.420  |10.852  |  0.336  | 0.692  | 0.869 |  0.934|
> |back | 0.196  |  3.560 | 14.276  | 0.300  |  0.734  |  0.897  |  0.951|
>
> Monodepth2:
> | View| Abs Rel | Sq Rel | RMSE | RMSE log | delta < 1.25 | delta < 1.25^2 | delta < 1.25^3 |
> |:-------:|:-------:|:-------:|:-------:|:-------:|:-------:|:-------:|:-------:|
> | front | 0.152 | 3.557  |13.898  | 0.241 |  0.826  | 0.933 |  0.968|
> |front left | 0.213 | 3.670  |12.161  | 0.315  | 0.719  | 0.887  |  0.942|
> |back left| 0.235 |  4.411 | 12.092  | 0.328  | 0.704  | 0.876  |  0.938|
> |front right| 0.239 |  3.846  | 11.630  | 0.337  | 0.677  | 0.867  |  0.932|
> |back_right|  0.244  |  3.781  | 11.142  |  0.345  | 0.677  |   0.858  | 0.930|
> |back | 0.218  | 4.222  |14.322 |  0.314  |  0.700 |   0.891 |   0.948|

---

> ### Author Response · Authors · 2022-08-26
> **Robust analysis**
>
> Dear viewer:
>
> we conducted robust analysis experiment on DDAD dataset about camera occlusion and camera delay. Since our method does not use extrinsics during inference, extrinsic noise will not affect inference performance. For camera occlusion, we use a black image to substitute occluded view. For camera delay, we use $t-1$ frame to substitute current $t$ frame. We perform camera occlusion or camera delay for one view in order. Then we collect all other (not performered) views' results. Experimental results show that slight  camera occlusion and camera delay do not frop models' performance since skip connection in CVT is robust to the change from other views.
>
> | type   | Abs Rel | Sq Rel |RMSE | RMSE log | delta < 1.25 | delta < 1.25^2 |  delta < 1.25^3 |
> |:-------:|:-------:|:-------:|:-------:|:-------:|:-------:|:-------:|:-------:|
> |camera occlusion| 0.199 | 3.387| 12.262| 0.301| 0.740| 0.895| 0.947|
> |camera delay| 0.198 | 3.327| 12.131| 0.300 | 0.742| 0.895| 0.947|
> |origin | 0.200| 3.392 |12.270 |0.301 |0.740 |0.894 |0.947|

---

### Official Review · Reviewer_CgbY · 2022-08-01

**Originality:** Good
**Technical Quality:** Very Good
**Clarity Of Presentation:** Good
**Impact:** 3

**Recommendation:**

Weak Accept: I recommend accepting the paper, but will not argue for my recommendation if the majority of other reviewers have a different opinion.

**Summary:**

This paper aims at a unique problem of multi-camera surrounding view self-supervised depth estimation. A cross-view transformer is designed and applied at multiple scales to incorporate multi-view features. Comprehensive experiments are conducted for depth evaluation.


**Issues:**

- The scientific novelty and method generality should be highlighted. Since the method cannot theoretically guarantee the multi-view depth consistency, the authors are suggested to construct a few research hypotheses for the scientific novelties.
- Joint pose estimation is included but not fully evaluated (preferred).

**Quality Of The Limitations Section:**

Additional details required

**Reviewer Expertise:**

4: The reviewer is confident but not absolutely certain that the evaluation is correct

**Robotics Focus:**

Highly relevant to robotics but no hardware experiments

**Strengths And Weaknesses:**

Strengths:
+ This research is well-motivated for multi-view cameras on autonomous vehicles.
+ This paper is generally well-written in a good structure.
+ This paper mostly achieves SOTA performance in DDAD and nuScenes datasets.

Weaknesses:
- The proposed method cannot theoretically guarantee the multi-view depth consistency.
- The computational performance is not evaluated for the robotic context.


**Summary Of Recommendation:**

This paper presents a novel cross-view transformer to fuse multi-camera features in a multi-scale fashion for a unique problem - multi-surrounding view camera depth estimation. The application is novel, although the methodology is still considered incremental. Given the performance in this unique problem, the review suggests 'Weak Accept' or slightly below that.

---

> ### Author Response · Authors · 2022-08-20
> **The scientific novelty and method generality**
>
> Dear reviewer:
>
> Thanks for your valuable comments. There are two main contriibution of our work: 1. We prove that integrate multi-view features with transformer can boost both the performance and depth consistency. 2. We propose SfM pretraining and joint pose estimation to recover real-world scales. Since the proposed cross-view transformers and SfM pretraining need overlaps to extract features and correspondences respectively, a key research hypothese is that there must have overlaps between multi-camera views. Moreover, from Table 1-4, we can find that our model performs better on DDAD than nuScenes. We notice that the overlap between multi-camera images in DDAD is larger than that in nuScenes and the experimental results indicate that our method prefers large overlap. This is reasonable since large overlap can benefit cross-view interaction and we can extract more correspondences and pseudo depths.

---

> ### Author Response · Authors · 2022-08-23
> **Joint pose estimation**
>
> Dear reviewer:
>
> To fully evaluate joint pose estimation, we evaluate temporal ego-motion errors on DDAD validation set. Specifically, we convert rotation matrix to Euler angles ($\phi$, $\theta$, $\psi$) and calculate Abs error between ground-truth Euler angles and predicted ones. The experimental result shows that except depth, we can also get more accurate ego-motion with the proposed joint pose estimation.
>
> | method    | $\phi$ error | $\theta$ error|$\psi$ error|
> |:-------:|:-------:|:-------:|:-------:|
> |separate pose|1.54e-3|6.27e-4|4.26e-4|
> |joint pose|8.29e-4|4.83e-4|4.70e-4|

---

### Meta-Review · Area_Chair_T2ev · 2022-08-14

**Recommendation:** Accept (Poster)
**Confidence:** 4

**Metareview:**

The paper received overall favorable reviews pre-rebuttal. The reviewers agree that the proposed method is novel, technically sounds, and performant. They also acknowledge the presentation quality. To improve the manuscript, the reviewers suggest that the author should (1) clarify assumptions regarding view overlaps (RZGD), (2) robust analysis (RZGD), (3) further clarify if using multiple views clearly improve the performance (maRz), (4) visualize result in point cloud (maRz), (5) make evaluation setting consistent with that of FSM (nGH3), and (6) discuss/evaluate the computation performance of the method in practical settings.

Post rebuttal: The authors have addressed majority of the concerns during the response period. The reviewers have all agreed that the paper studies an important problem and provides a convincing solution, provided that the paper will be updated with the following content: (1) comparison with FSM, (2) justification of multi-view performance gain, (3) open source code, and (4) detailed analysis of runtime performance. The paper is recommended to be accepted and presented as a poster.

**Best Paper Nomination:**

No

---

> ### Author Response · Authors · 2022-08-26
> **Responce to reviewers' questions**
>
> Dear area chair:
>
> Thanks for your detailed summary. For now, we have answered all viewers' questions and we will revise our paper according to reviewers' suggestions. It is a delight that we have addressed the concerns of Reviewer maRz whose initial score is "strong reject", and the reviewer agreed to change the score!